# Comparative multi-omics analyses reveal differential expression of key genes relevant for parasitism between non-encapsulated and encapsulated *Trichinella*

Xiaolei Liu[1,5], Yayan Feng [2,5], Xue Bai[1], Xuelin Wang[1], Rui Qin[2], Bin Tang[1], Xinxin Yu[2], Yong Yang[1], Mingyuan Liu [1,3✉] & Fei Gao [2,4✉]

Genome assemblies provide a powerful basis of comparative multi-omics analyses that offer insight into parasite pathogenicity, host-parasite interactions, and invasion biology. As a unique intracellular nematode, *Trichinella* consists of two clades, encapsulated and non-encapsulated. Genomic correlation of the distinct differences between the two clades is still unclear. Here, we report an annotated draft reference genome of non-encapsulated *Trichinella*, *T. pseudospiralis*, and perform comparative multi-omics analyses with encapsulated *T. spiralis*. Genome and methylome analyses indicate that, during *Trichinella* evolution, the two clades of *Trichinella* exhibit differential expansion and methylation of parasitism-related multi-copy gene families, especially for the DNase II members of the phospholipase D superfamily and Glutathione S-transferases. Further, methylome and transcriptome analyses revealed divergent key excretory/secretory (E/S) genes between the two clades. Among these key E/S genes, TP12446 is significantly more expressed across three life stages in *T. pseudospiralis*. Overexpression of TP12446 in the mouse C2C12 skeletal muscle cell line could induce inhibition of myotube formation and differentiation, further indicating its key role in parasitism of *T. pseudospiralis*. This multi-omics study provides a foundation for further elucidation of the mechanism of nurse cell formation and immunoevasion, as well as the identification of pharmacological and diagnostic targets of trichinellosis.

[1] Key Laboratory of Zoonosis Research, Ministry of Education, Institute of Zoonosis/College of Veterinary Medicine, Jilin University, Changchun, China. [2] Shenzhen Branch, Guangdong Laboratory for Lingnan Modern Agriculture, Genome Analysis Laboratory of the Ministry of Agriculture, Agricultural Genomics Institute at Shenzhen, Chinese Academy of Agricultural Sciences, Shenzhen, China. [3] Jiangsu Co-innovation Center for Prevention and Control of Important Animal Infectious Diseases and Zoonoses, Yangzhou, Jiangsu, PR China. [4] Comparative Pediatrics and Nutrition, Department of Veterinary and Animal Sciences, Faculty of Health and Medical Sciences, University of Copenhagen, Frederiksberg DK, Denmark. [5]These authors contributed equally: Xiaolei Liu, Yayan Feng. ✉email: liumy36@163.com; flys828@gmail.com

Trichinellosis is a worldwide zoonotic disease, caused by an intracellular nematode of the *Trichinella* genus, which causes substantial morbidity and mortality in both animals and humans. Molecular studies have identified 12 species and genotypes that parasitize a wide range of vertebrate hosts[1]. These *Trichinella* spp. are categorized of two principal evolutionary clades, the encapsulated represented by *Trichinella spiralis* and non-encapsulated by *Trichinella pseudospiralis*. Both clades have same life cycles occupying two distinct intracellular niches, intestinal epithelium and skeletal muscle cell. The muscle larvae (ML) of *Trichinella*, released by host gastric fluids, invade intestine epithelium and subsequently develop into adult worms (Ad). The newborn larvae (NBL) delivered by female adults migrate to skeletal muscles and invade into muscle cells where they develop into ML and survive for years[2].

The difference in parasitological, pathological, and immunological characteristics between *T. pseudospiralis* and *T. spiralis* has been an interesting topic[3–5]. During the formation of niches for long term parasitism, differences in activation, proliferation and fusion of satellites, regeneration and degeneration of cytoplasm[6–8], re-differentiation, apoptosis and transformation of the ML infected muscle cells[9] were revealed. These differences lead to the consequence of formation of two types of niches, typical nurse cells in *T. spiralis* that is characterized by a thick surrounding collagen capsule, and non-typical nurse cells in *T. pseudospiralis* that is characterized by poorly developed collagen capsule, slow disintegrated amorphous cytoplasm, and continuous and diffuse myopathy in whole length of the infected muscle fiber[10]. The differences between *T. pseudospiralis* and *T. spiralis* also extend to aspects of the host immune and inflammatory responses to infection. *T. pseudospiralis* is less pathogenic than *T. spiralis*, inducing considerably less inflammation in the intestine and muscles of hosts[11,12]. Studies have been tried to reveal the mechanism of these differences between the two clades of *Trichinella*, but so far it is still unclear. One of most considered inducers is the E/S secreted by Ad and ML stages, which is proposed to play important roles in nurse cell formation and immune responses[13,14].

Previous studies implied a substantial difference in the genomes of encapsulated compared with non-encapsulated taxa of *Trichinella*, based on phylogeny analysis[15]. Comparative analysis at both genome and transcriptional regulation levels between the representative encapsulated and non-encapsulated clades will help to elucidate the parasitism mechanism. However, until recently, a high-quality draft genome assembly was only available for *T. spiralis*[16]. The published *T. pseudospiralis* genome assembly was constructed from short-read shotgun sequencing, resulting with a big gap between the assembly and the estimated genome size by flow cytometry[17].

Here we present a high-quality long-read-assembled reference genome of *T. pseudospiralis*, in which the majority of the repeat regions have now been assembled. This was achieved by coupling single-molecule real-time (SMRT) sequencing (Pacific Biosciences) with Illumina sequencing technologies to assemble a significantly improved version of the *T. pseudospiralis* genome. Based on this assembly of high-quality reference genome, characterization of DNA methylation machinery, DNA methylomes and transcriptomes across three life stages of *T. pseudospiralis* was then performed. Further, multi-omics analyses were applied to comprehensively address the molecular differences of *T. spiralis* and *T. pseudospiralis*. Thereby, we revealed extensive differences of repetitive genome extent, DNA methylation levels as well as gene expression program between the two clades of *Trichinella* species. Furthermore, we identified sets of excretory/secretory (E/S) genes, whose DNA methylation and expression level varied significantly between the two species. Functional study on one of the *T. pseudospiralis*-specific highly expressed E/S genes, TP12446, further

demonstrated its key role in parasitism by inducing inhibition of myotube formation and differentiation of C2C12 cell line.

## Results

**Genome assembly and features.** We used one non-encapsulated strain (ISS13) for de novo assembly of *T. pseudospiralis* reference genome, using a 'hybrid' approach that combined assembly of PacBio and Illumina reads. The average read depth of PacBio and Illumina reads were 144× and 98×, respectively (Supplementary Table 1). After error correction, the total length of the assembled genome was 68.90 Mb, which represents 98.7% of the genome size, estimated by $k$-mer depth distribution of the sequenced Illumina reads (69.79 Mb; Supplementary Fig. 1). The assembly consists of 2746 scaffolds (≥500 bp) with a mean GC-content of 31.49% and N50 lengths of 208.90 Kb, among which 68 of the largest scaffolds spanning more than half of the *T. pseudospiralis* genome (Table 1). The quality of the assembly was further confirmed using available RNA-seq (Supplementary Table 2) and expressed sequence tag (EST) sequences (Supplementary Table 3). Then we used Benchmarking Universal Single-Copy Orthologs (BUSCO) to assess the completeness of the genome based on the presence of single-copy orthologs from the OrthoDB database. We found that 870 (88.6%) out of 982 genes (nematoda_odb9 lineage) were present and complete in the present study (hereafter named T4_ISS13_R), which are comparable with that of the well-assembled *T. spiralis* genome, as 876 (89.2%) out of 982 genes were identified and complete in the *T. spiralis* genome (Table 1). In the newly generated version of *T. pseudospiralis* reference genome, a total of 27.76 Mb of non-redundant repetitive elements were identified, which represents ~40.28% of the genome. Approximately 86.5% of these repeat regions were well-assembled based on high coverage folds (≥150×). Then we compared the two assembly versions of *T. pseudospiralis*. In addition to the comparable gene features of T4_ISS13_R in comparison to previously generated assembly version T4_ISS13_r1.0, the T4_ISS13_R also had approx. 18.9 Mb more repeat sequences than T4_ISS13_r1.0, accounting for most of the differences in genome size between the two assembly versions (Supplementary Table 4). Thus, this assembly T4_ISS13_R was used for further analyses.

In T4_ISS13_R, we predicted 12,682 protein-coding genes (Supplementary Table 5), spanning 19.1% of the T4_ISS13_R genome. The mean length of all the predicted genes is 2.33 Kb, with an average of 5.5 exons and a mean exon length of 188.4 bp

**Table 1 Comparison of genome assembly and annotation between *T. spiralis* and *T. pseudospiralis*.**

| Description | *T. spiralis* | *T. pseudospiralis* |
|---|---|---|
| Total scaffolded assembly size (Mb); total scaffolds | 63.53; 6863 | 68.90; 2,746 |
| Total scaffolds of >2 kb: length (Mb); no. of scaffolds | 57.27; 964 | 68.71; 2,631 |
| Largest scaffold (Mb) | 12.04 | 3.73 |
| N50 in kb of scaffolds; count > N50 length | 6373.44; 4 | 208.90; 68 |
| N90 in kb of scaffolds; count > N90 length | 2.05; 919 | 7.68; 1,231 |
| GC content of the whole-genome (%) | 33.9 | 31.49 |
| Number of gene models | 16,186 | 12,682 |
| Mean/Median gene size (bp) | 1837.73/1096 | 2330.43/1077 |
| Mean number of exons per gene | 5.41 | 5.52 |
| Mean/Median exon size (bp) | 178.45/129 | 188.37/137 |
| Repetitive sequences (%) | 20.66 | 40.28 |
| BUSCO genes (%) | 89.2% | 88.6% |

per gene (Table 1). Approximately 89.17% of the genes had either known homologs or could be functionally classified (Supplementary Table 6). In addition, we also annotated potential parasitism-related functional proteins, such as proteinases, protein kinases, G protein-coupled receptors (GPCRs), and excretory/secretory (E/S) proteins, as well as the candidate molecular targets for treatment of trichinellosis. To this end, we have identified 1627 proteinases, 88 GPCRs, 336 kinases, 471 E/S proteins and 154 potential drug targets in *T. pseudospiralis*. A similar number of functional proteins and candidate drug targets were also annotated in the *T. spiralis* genome (Supplementary Data 1). Moreover, approx. 93.0–95.6% of these proteins had support of gene expression data, and could act as important factors in immuno-evasion and excystment/encystment, as suggested by the previous studies[18].

**Methylome annotation and comparison.** Previously, we confirmed the existence of DNA methylation in the *T. spiralis* genome and its relevance to parasitism[19]. To further annotate the genome regulation of *T. pseudospiralis*, we applied both bioinformatics analyses of homologous gene sequences and enzymatic tests to confirm the existence and activity of DNA-methyltransferase 1 (DNMT1) and DNMT3 in *T. pseudospiralis* (Fig. 1a; Supplementary Fig. 2). Single-base resolution maps of DNA methylation for three life stages were then generated using whole-genome bisulfite sequencing (Supplementary Table 7). Similar to the *T. spiralis* genome, the *T. pseudospiralis* genome also displayed stage-specific methylome patterns, in which Ad and ML stages were moderately methylated, while NBL stage showed rare traces of DNA methylation (Supplementary Figs. 3 and 4), both for genic regions and repetitive sequences, across the entire genome (Fig. 1b; Supplementary Data 2 and 3). Between the two genomes, a clear genome-wide methylation difference of non-repetitive regions was observed, as indicated by hierarchical clustering analysis (Fig. 1c; Supplementary Data 2–5). More specifically, *T. pseudospiralis* presented a heavier global methylation level than *T. spiralis* in the non-repetitive regions (Fig. 1d; Supplementary Data 2–5). Such higher methylation within the *T. pseudospiralis* genome was not limited to specific region but were broadly distributed in genic regions, intergenic regions and different repetitive sequences across the genome, except for intron regions (Supplementary Fig. 5) and long interspersed nuclear elements (LINEs) (Supplementary Fig. 6). These results are in accordance with the CpG content (CpG observed/expected [o/e]) for a total of 1436 pairs of ortholog single genes being lower in *T. pseudospiralis* than in *T. spiralis*, both across the genome and coding regions (Supplementary Fig. 7). The depletion of normalized CpG o/e values may represent an evolutionary signature of DNA methylation in animal genomes, as methylated cytosines undergo spontaneous deamination to thymine with high frequency.

Furthermore, we quantified the epigenetic conservation level using the *P*-value generated by comparing the common methylated CpGs of orthologous genes between the two species. We found that no clear correlation was observed between the methylation and the gene sequence differences (Fig. 1e; Supplementary Data 2 and 3); however, a much higher correlation was observed between epigenetic and transcriptional divergences (Fig. 1e; Supplementary Data 6). These results coincide with previous studies that epigenomic conservation is not a simple consequence of sequence similarity, but rather a regulatory mechanism for transcription[20].

**Comparative genomics.** We next evaluated organizational characteristics of the genomes of *T. pseudospiralis* and *T. spiralis*. The number of predicted genes in *T. pseudospiralis* ($N = 12,682$) is notably lower than the 16,186 genes identified in *T. spiralis*, which gave a higher gene density in *T. spiralis* (254 per Mb in *T. spiralis* and 184 per Mb in *T. pseudospiralis*; Student's *t*-test $P < 2.2e-16$), even though the two genomes show a similar genome size (63.53 Mb in *T. spiralis* and 68.90 Mb in *T. pseudospiralis*). A comparison of 6857 orthologous genes (based on reciprocal best BLAST hits) in the *T. pseudospiralis* and *T. spiralis* genomes indicated that *T. pseudospiralis* has a significantly longer average intron size compared to *T. spiralis* (240 bp compared to 177 bp; Student's *t*-test $P < 2.2e-16$), whereas the average exon size is relatively similar for the two species (199 bp for *T. pseudospiralis* and 194 bp for *T. spiralis*; Student's *t*-test $P = 0.08$). A clear correlation was observed between the length of repeats in the introns and intron length, per se, in *T. pseudospiralis*, thus indicating that repeats contributed to the greater intron length of *T. pseudospiralis* compared to that in the *T. spiralis* genome (Supplementary Fig. 8). As the most obvious difference between the two genomes lay within the repeat content, we next focused on the repeat level of different categories between the two genomes. Among the various categories, the *T. pseudospiralis* genome contained obviously elevated levels of both tandem repeats (11.51 Mb in *T. pseudospiralis* and 717.76 Kb in *T. spiralis*) and long terminal repeats (LTRs) (15.31 Mb in *T. pseudospiralis* and 4.33 Mb in *T. spiralis*) compared to the *T. spiralis* genome (Supplementary Table 8). After calculating for insertion times, we discovered that a burst of LTR activity likely occurred during the last five million years (Supplementary Fig. 9). By comparing with species phylogeny, an event estimated to have occurred 22.6 (15.3–28.1) million years ago (Supplementary Fig. 10), we deduced that a burst of LTR insertion were likely occurred after the divergence of the encapsulated and non-encapsulated *Trichinella*. In addition, the percentage of de novo predicted repeats was notably higher than that obtained from homologous predictions in *T. pseudospiralis*, based on Repbase, indicating that *T. pseudospiralis* has many unique repeats compared to *T. spiralis* genome (Supplementary Table 8).

A Markov clustering algorithm was adopted to delineate gene family expansion and contraction events. The analysis included species from four major phylogenetically-related lineages that collectively span the phylum, including two *Trichinella* species (*T. spiralis* and *T. pseudospiralis*), one non-parasite *C. elegans*, one plant parasite, *M. incognita*, one animal parasite, *B. malayi*, thus representing diverse trophic ecologies. Arthropod (*D. melanogaster*) was used as outgroup. Accordingly, a total of 14,749 orthologous gene families were generated from five nematode species. Based on the comparison of orthologous gene families, the *T. pseudospiralis* genome displayed 628 expanded and 1087 contracted gene families, compared with the *Trichinella* common ancestor. By contrast, the *T. spiralis* genome displayed more expanded (1376) than contracted (884) ones (Fig. 2a; Supplementary Data 7). Among the five species of nematodes, 2543 families are broadly conserved (NOG), whereas 337 families (containing 1818 genes) or 684 families (containing 2577 genes) appear to be *T. pseudospiralis* or *T. spiralis* specific, respectively (Fig. 2b).

Collectively, our results here revealed that the two clades of *Trichinella* showed substantial differences in relation to genome features, especially for repeat contents and gene family expansion and contraction events.

**Differential expansion of gene families between the two clades of *Trichinella*.** Further, we observed a significantly higher level of TE density in the expanded gene families than the non-expanded ones, both in *T. pseudospiralis* (Student's *t*-test $P < 2.2e-16$) and *T. spiralis* (Student's *t*-test $P < 2.2e-16$) (Supplementary Fig. 11a). In association with TE density, DNA methylation levels were also

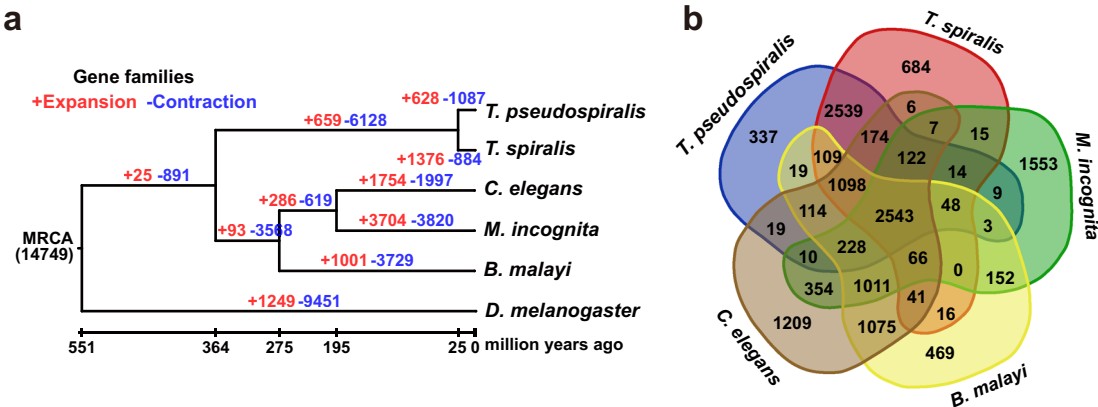

**Fig. 1 Confirmation and characterization of *T. pseudospiralis* methylome in comparison with *T. spiralis*. a** Results of catalytic activity of *T. pseudospiralis* DNMTs. Triplicates of DNMT activity experiments were carried out and mean ± SD is indicated. DNMT activity (OD/h/mg) = (Sample OD−Blank OD)/ (Protein amount (μg) × hour) (N = 3 biologically independent experiments for each DNMT). **b** CpG methylation levels of repeat and gene body regions. Two-kilobase region upstream and downstream of each gene was divided into 100-bp (bp) intervals. Each repeat or gene body region was divided into 20 intervals (5% per interval). **c** Clustering of methylation levels of common CpG sites in the whole-genome of all the four samples was used in the "Pvclust" algorithm. **d** Comparison of mCG methylation level density between *T. pseudospiralis* and *T. spiralis*. **e** Correlations among evolutionary changes of epi-modification intensities, gene expression levels, and genomic sequences.

**Fig. 2 Comparative genomics between *T. pseudospiralis* and *T. spiralis*. a** The dynamics of gene family sizes in the genomes of parasitic nematode *T. pseudospiralis*, *T. spiralis*, *C. elegans*, *M. incognita*, *B. malayi*, with *D. melanogaster* serving as the outgroup. Numbers above the branches represent gene family expansion and contraction events, respectively. The plus sign represents expansion events and the minus sign indicates contraction events. **b** Number of shared gene families between *T. pseudospiralis* and four other nematodes (that is, *T. spiralis*, *C. elegans*, *M. incognita*, and *B. malayi*).

highly elevated in these expanded gene families (Supplementary Fig. 11b). To characterize the association among TE density, DNA methylation and expansion of gene families, we further defined the TE-enriched gene families as those that contain significantly higher TE density in gene-body or 2-Kb flanking regions of transcriptional start sites (TSSs) relative to the genome overall (Student's $t$-test $P < 0.05$). As a result, we identified 282 and 380 TE-enriched gene families in *T. pseudospiralis* and *T. spiralis* genomes, respectively. 67 and 183 gene families were significantly expanded in *T. pseudospiralis* and *T. spiralis*, respectively (Fisher's exact test $P < 0.05$). Most of these gene families contained domains without known functional annotations, while only a small number of gene families contained unitary functional domains based on IPR annotations ($N = 4$ in *T. pseudospiralis* and $N = 14$ in *T. spiralis*, Supplementary Table 9).

Among these gene families, *T. spiralis* showed more expansion of Deoxyribonuclease II (DNase II) family (IPR004947, $N = 43$ in *T. pseudospiralis* versus $N = 133$ in *T. spiralis*) and unknown function DUF1759 family (IPR005312, $N = 22$ in *T. pseudospiralis* versus $N = 103$ in *T. spiralis*) (Supplementary Fig. 12). By reconciling the species phylogeny (Supplementary Fig. 10) with the gene phylogeny and by genomic/scaffold location of the DNase II genes, we observed that only part, but not all of the family members underwent an expansion (Fig. 3a). More specifically, 8 families containing 15 genes in the *T. pseudospiralis* genome were expanded into 57 genes in the *T. spiralis* genome, but no *T. spiralis* gene families were expanded in *T. pseudospiralis* (Fig. 3b; Supplementary Table 10). Then we observed that TE density of the expanded DNase II loci in *T. spiralis* is ~1.5 times higher than that of the entire genome (Supplementary Fig. 13), meanwhile significantly higher than that in *T. pseudospiralis* (Student's $t$-test $P < 0.05$; Fig. 3c). Accordingly, we observed significantly higher methylation level of the expanded genes than that of the non-expanded genes in gene-body or promoter regions in *T. spiralis* (Mann–Whitney $U$ test $P < 0.05$; Fig. 3d; Supplementary Fig. 14; Supplementary Data 4 and 5), along with negatively correlated gene expression levels both in Ad and ML stages (Fig. 3e; Supplementary Data 8).

In contrast, *T. pseudospiralis* showed significantly more expansion of TB2/DP1/HVA22-related protein family (IPR004345, $N = 61$ in *T. pseudospiralis* versus $N = 18$ in *T. spiralis*) and glutathione S-transferases (GSTs) family (IPR004046, $N = 13$ versus $N = 7$) (Fisher's exact test $P < 0.05$; Supplementary Table 9). Phylogenetic analysis revealed that apart from the known GSTs that showed homologous with GSTs in *T. spiralis* (TP07287 and TP02356), *T. pseudospiralis* additionally expanded one gene family (Fam1884) (Fig. 4a). Most of the genes in Fam1884 were well-assembled based on high coverage folds (90×–150×) and had an average of 9 potential antigenic epitopes predicted by ABCpred server using artificial neural network[21]. As the GSTs family are related with invasion and migration of *Trichinella*[22], its expansion might play an important role in the parasitism of *T. pseudospiralis*. Different from the DNase II genes in *T. spiralis*, a majority of the expanded GST genes in *T. pseudospiralis* showed approximately 2.0× higher TE density of the upstream and downstream regions in comparison to *T. spiralis* (Fig. 4b). Likewise, DNA methylation also involved in the regulation of duplicated GSTs in *T. pseudospiralis*, as higher methylation levels (Fig. 4c; Supplementary Data 2 and 3) and lower average expression levels were observed in *T. pseudospiralis* (Fig. 4d; Supplementary Data 6). Collectively, these results suggest that expanded gene families enriched with transposons and DNA methylation, such as DNase II and GSTs families, may potentially regulate differential parasitism between the two clades of *Trichinella*.

**Divergent E/S genes in relation with varied parasitism in *Trichinella*.** The E/S proteins are central to understand host-parasite

interactions or host cell modification[13]. Among the 471 E/S genes we identified in *T. pseudospiralis*, a large proportion were gene families that are highly divergent from *T. spiralis*, including TE-enriched DNase II, and TB2/DP1/HVA22-related proteins (Supplementary Data 9) as well as the non-TE-enriched C-type lectin gene family. In the present study, we identified nine C-type lectins (or lectin-like) E/S genes in *T. pseudospiralis*, but only a single gene in *T. spiralis*. Seven of these lectin genes were supported by RNA-seq data in *T. pseudospiralis*, thereby providing a more richer expression pool compared to *T. spiralis* (Fig. 5a; Supplementary Data 6 and 8). Six of the *T. pseudospiralis* lectins showed more homology to the *T. spiralis* lectin than to *C. elegans* lectin genes, whereas the other three *T. pseudospiralis* lectins shared a greater level of identity with mammalian C-type lectin domains (Fig. 5b). Sequence alignment of the three proteins with their mammalian lectin homologs indicated that the key cysteine residues were highly conserved in *T. pseudospiralis* lectin proteins, along with additional residues implicated in forming the stable hydrophobic protein core (Supplementary Fig. 15; Supplementary Table 11). Structure modeling predicted that TP12499 displayed a remarkable structural similarity to the mammalian immune-system lectin Mincle in the PDB library (Supplementary Fig. 16), which can induce acquired immunity, such as antigen-specific T-cell responses and antibody production[23]. Thus, it may form stable complexes with galactose, as suggested by the high C-score of the models (C-score = 0.46). In addition, TP02221 exhibited high structural similarity with MRC2. Hence, it may also bind with mannopyranoside (C-score = 0.44) to implement cell infection through endocytosis and this then contributes to antigen presentation (Supplementary Fig. 16). Finally, TP10242 shared a similar structure with Dectin-2 and may, therefore, function through a raffinose binding site (C-score = 0.17) (Supplementary Fig. 16; Supplementary Table 11). Thereby, through these expanded C-type lectin superfamily members, *T. pseudospiralis* might have developed an enhanced capacity for immune evasion compared to *T. spiralis*.

In addition, we addressed stage-specifically expressed genes between the two species and identified 112 and 91 stage-specific single-copy orthologous (SCO) E/S genes in *T. pseudospiralis* and *T. spiralis*, respectively (Fig. 5c). Among these SCO E/S genes, 50 genes were homologous between the two genomes (Fig. 5d). 34 out of the 50 genes were differentially expressed between the two species (Fisher's exact test $P < 0.01$) (Supplementary Fig. 17). Of note, 6 genes showed species-specific divergence in at least two stages of *T. pseudospiralis* (Supplementary Table 12). Among these genes, TP12446, annotated as E3 ubiquitin-protein ligase RNF128, was a transcription factor with a zinc-finger (RING finger) motif at its C-terminal region. Expression analysis revealed that TP12446 was a ML stage-specifically expressed gene in *T. pseudospiralis* and was significantly upregulated across three life stages in comparison to *T. spiralis* (Fisher's exact test $P < 0.05$; Supplementary Table 12). We speculated this upregulation was modulated by DNA methylation as significantly higher methylation level was observed in gene-body or promoter regions of *T. pseudospiralis* in comparison to *T. spiralis* (Mann–Whitney $U$ test $P < 0.05$; Supplementary Fig. 18). Structural modeling revealed that TP12446 shared protein sequence and structure similarity with human E3 ubiquitin-protein ligase, RNF38 (homology modeling TM-score = 0.844) (Fig. 5e). Based on these findings, we speculated that TP12446 may function at host-parasite interface.

**Secreted TP12446 induced cytoskeleton disarrangement.** Further, we employed mouse C2C12 cell line as a model to evaluate the potential regulation on host cells by the secreted TP12446 from

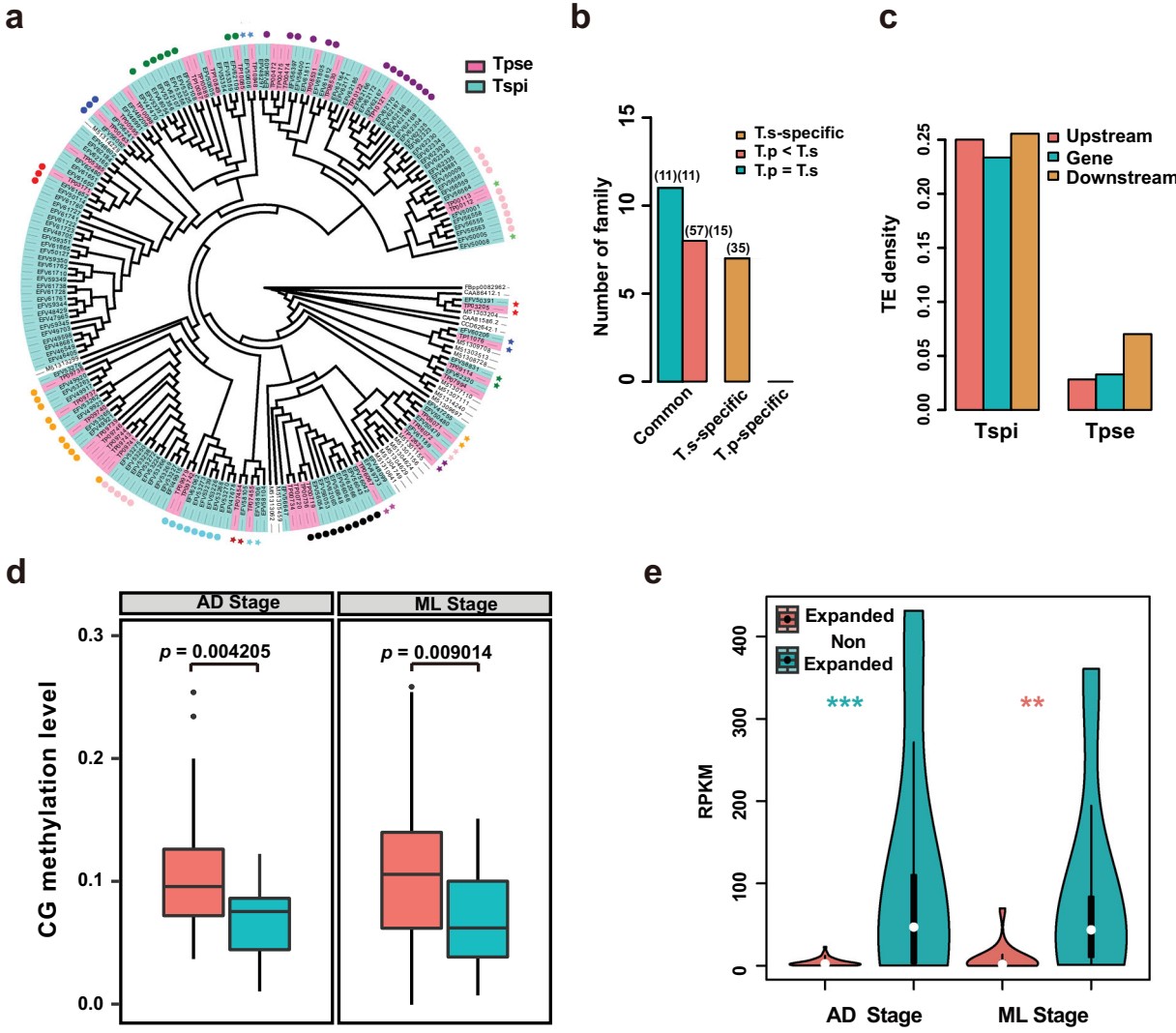

**Fig. 3 Differential expansion of DNase II gene family between the two clades _Trichinella_. a** Phylogenetic tree of DNase II gene family with _T. pseudospiralis_ (colored in pink), _T. spiralis_ (colored in cyan) and other three species (that is _D. melanogaster_, _C. elegans_ and _T. suis_). **b** Common and specific number of DNase II family between _T. pseudospiralis_ and _T. spiralis_. Numbers in the front and back brackets represent gene number and family number, respectively. **c** Comparison of the average density of transposon element (y-axis) around DNase II loci and their two-kilobase flanking regions upstream and downstream in _T. pseudospiralis_ and _T. spiralis_ (x-axis). **d** Comparison of methylation level between expanded (red) and non-expanded (green) DNase II genes in Ad (left) and ML (right) stages in _T. spiralis_. P values were calculated using Mann–Whitney _U_ test. **e** Comparison of expression level between expanded (red) and non-expanded (green) DNase II genes in Ad and ML stages in _T. spiralis_. A Student's _t_-test was applied to the pairwise comparison. *** indicates _P_ < 0.001; ** indicates _P_ < 0.01.

_T. pseudospiralis._ Sequence of TP12446 was confirmed by PCR analysis and cloned with a FLAG epitope tag into a lentiviral vector. The mouse C2C12 cells were first differentiated into myotubes (Supplementary Fig. 19), considering _Trichinella_ reprograms terminally differentiated skeletal muscle cells. Then, we produced three transgenic C2C12 cell lines: (1) normal cell (Blank), (2) an empty lentiviral vector PSE-CMV-NC (NC), 3) a lentiviral vector PSE-CMV-TP12446 with overexpression of TP12466 (OE). The transduction efficiency was assessed by western bolt, which showed high protein expression of TP12446 in PSE-CMV-TP12446 group both at day 4 and 8 when comparing with normal cell and PSE-CMV-NC groups (Supplementary Fig. 20). The myotube differentiation was further assessed by immunofluorescence analysis by examining the expression of the major histocompatibility complex (MHC), a myogenic differentiation marker. Normal myoblast exhibited short myotubes after 4 day of differentiation, which became more elongated with further culturing. For myotubes expressing TP12446, we

observed obviously morphological distinction from the empty vector control at day 4 and 8. Myotubes formed by control groups (Blank and NC) were numerous, long and often branched, and showed many nuclei distributed along their extension. On the contrary, myotube formation in TP12446 treated myoblasts was blocked and a sharp decrease in the number of MHC-expressing cells were observed (Fig. 6). Moreover, the myotubes generated in treated group were also shorter and thinner. These results indicated that TP12446 secreted by _T. pseudospiralis_ had significant impact on perturbation of myoblast formation and differentiation in vitro.

### Discussion

Comparative genomic and epigenomic studies require high-quality annotated reference genomes. For the zoonotic parasite _Trichinella_ spp., the first high-quality _T. spiralis_ reference genome, based on Sanger sequencing, was released in 2011[16]. Further, in 2016, assemblies of reference genomes for an

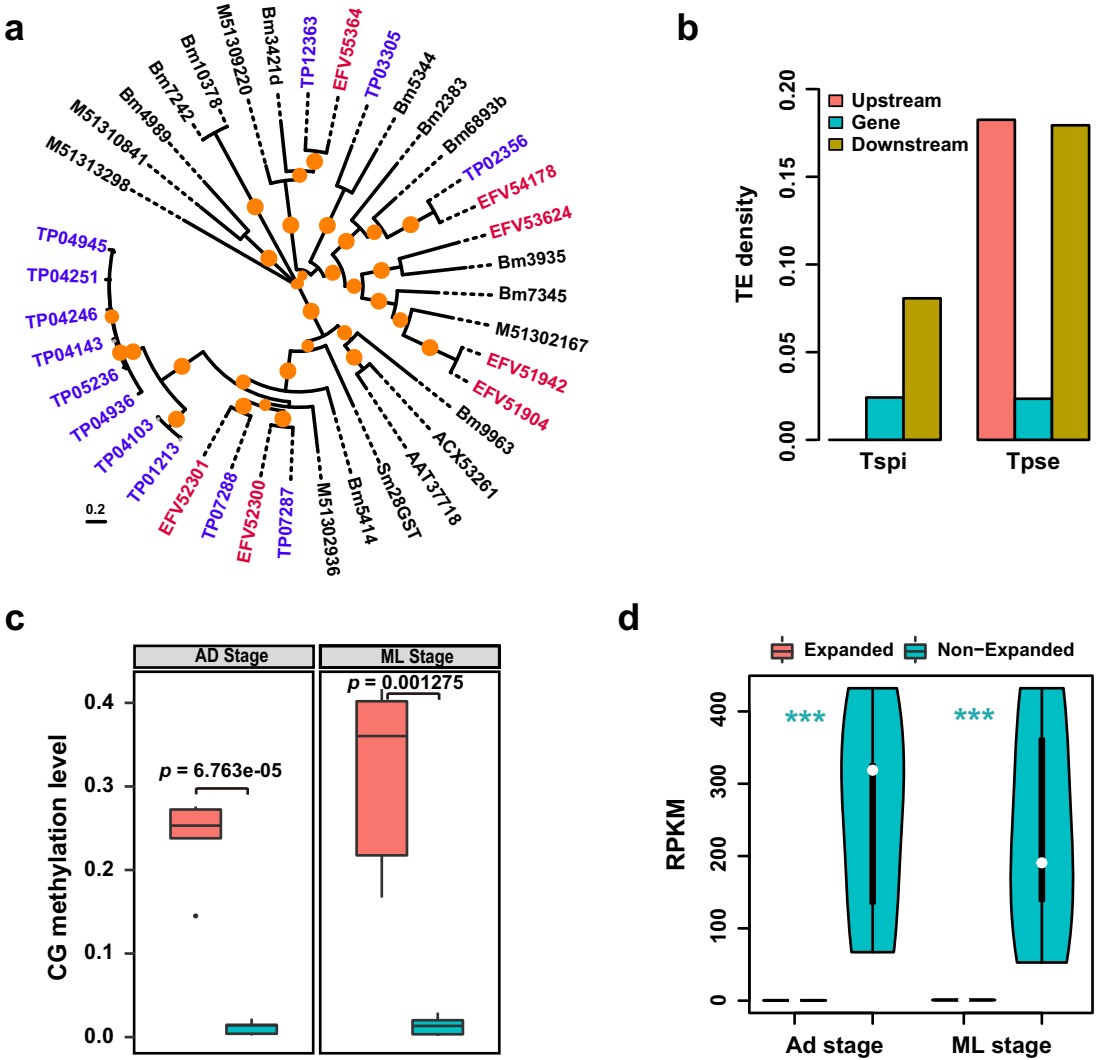

**Fig. 4 Differential expansion of glutathione S-transferases family between the two clades of *Trichinella*. a** Phylogenetic relationship of GSTs in *T. pseudospiralis*, other nematodes (that is *T. spiralis*, *B. malayi* and *T. sui*) and GSTs with experimentally verified (that is Sm28GST, ACX53261, AAT37718, EFV54178, and EFV52300). Gene names in blue are from *T. pseudospiralis* and those in red are from *T. spiralis*. Nodes with >30% bootstrap support (1000 replicates) are indicated in orange circles. **b** Comparison of average density of transposon element (y-axis) around GSTs and their two-kilobase flanking regions upstream and downstream in *T. pseudospiralis* and *T. spiralis* (x-axis). **c** Comparison of methylation level between expanded (red) and non-expanded (green) GSTs in Ad (left) and ML (right) stages in *T. pseudospiralis*. P values were calculated using Mann–Whitney U test. **d** Comparison of expression level between expanded (red) and non-expanded (green) GSTs in Ad and ML stages in *T. pseudospiralis*. A Student's t-test was applied to the pairwise comparison. *** indicates P < 0.001.

additional twelve *Trichinella* spp., including *T. pseudospiralis*, were published, applying next-generation sequencing technologies[15]. However, a large proportion of the published *T. pseudospiralis* genome was missing, as only 45 Mb of the genome was assembled, which is much less than 70 Mb of the genome size of *Trichinella* spp[17]. In order to study the molecular basis of parasitism, pathology and immune response for the encapsulated species (*T. spiralis*) and non-encapsulated species (*T. pseudospiralis*), high-quality *T. pseudospiralis* reference genome is required for comparative genome analyses.

In the present study, we have assembled a new *T. pseudospiralis* reference genome with large amounts of repeat sequences assembled and performed multi-omics analyses based on the newly annotated genome sequences. Comparative analyses revealed that the most significant differences between the two genomes lay within the repeat content, as 27.7 Mb and 13.1 Mb repeat sequences were identified in *T. pseudospiralis* and *T. spiralis* genomes, respectively. Stepwise buildup of repetitive

sequences not only has implications on genome stability, but also is considered as a transcription regulatory mechanism among distantly related species[24]. Thus *T. pseudospiralis* also showed higher methylation level than *T. spiralis* at genome-wide scale. Thereby, the two *Trichinella* parasites differ significantly from each other both at genomic and epigenomic level, forming the foundations for differential parasitism between the two *Trichinella* species.

As quantitative changes in gene family members can reflect molecular determinants underlying species adaptation and evolution, here, we also observed that the two clades of *Trichinella* exhibited significant differences with regard to gene family expansion and contraction events by comparative genomic analysis. Notably, we observed DNase II family was highly expanded in *Trichinella*, especially in *T. spiralis*. Considering the biological processes DNase II genes involved in, including penetration of host tissues, formation of nurse cells, and evasion of host immune response[25], such an extensive repertoire of DNase II in

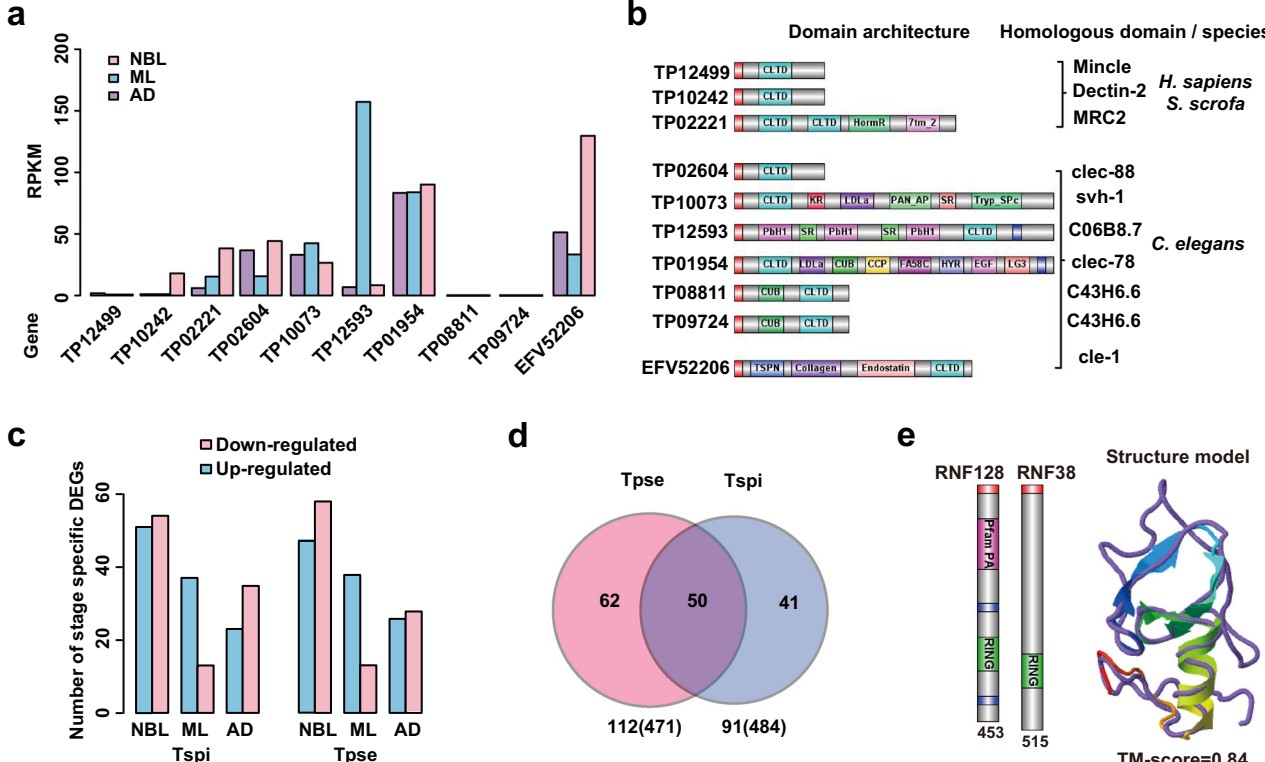

**Fig. 5 Stage-specific expressed E/S genes form the basis of varied parasitism between *T. pseudospiralis* and *T. spiralis*. a** The histogram shows expression levels of secreted C-type lectin superfamily across three life stages. **b** Domain architecture (left) and homologous domain sequences (right) in mammalian host of C-type lectin superfamily. **c** The histogram shows numbers of stage-specifically differential expressed genes (DEGs) between the two species across three life stages. **d** Venn diagram of homologous SCOs between total SCOs identified in the two species. **e** Domain architectures and structural model of TP12446 and RNF38. The subgraph numbers indicate the protein length. Query structure is shown in cartoon, while the structural analog in the PDB library is displayed using backbone trace and colored in purple (as identified by TM-align). The TM-score value scales the structural similarity of the two structures.

*T. spiralis* is of particular importance for the intracellular parasitic mechanisms, especially for the nurse cell formation that is different from *T. pseudospiralis* in activation, proliferation and fusion of satellite cell, re-differentiation and transformation of infected muscle cells[6]. In contrast, to adapt to the non-encapsulated phenotype, *T. pseudospiralis* showed significant expansion of GSTs family in comparison to *T. spiralis*. Previous studies have demonstrated that GSTs were likely related with cellular defense against toxic effects of metabolites, and the larval invasion and migration into intestinal epithelial cells (IECs), and vaccination with GSTs induced a low protective immunity against *T. spiralis* infection[26,27]. This higher expansion of GSTs in *T. pseudospiralis* might endow *T. pseudospiralis* enhanced invasion and migration and contribute to protective immunity against infection in order to adapt to its non-encapsulated phenotype. Hence, species-specific expansion of GSTs in *T. pseudospiralis* appeared to be potential targets against parasite infection.

Furthermore, expanded gene families of the two clades of *Trichinella* also showed increased level of TE density. Evidence in a variety of species have proved that transposon elements could potentially facilitate different types of segmental duplication based on their density and distribution, such as in odorant receptor loci (OR) of clonal raider ant and in *D. melanogaster*[28,29]. The increased TE density of expanded gene families might suggest the potential mechanism involved in the genomic architecture and organization. Moreover, we also observed elevated methylation level in expanded gene families than non-expanded ones. It has been argued that DNA methylation levels in *Trichinella* proved to be a mechanism for

life cycle transition[19]. Although our study shed light on the association between TE density and DNA methylation in the expanded gene families that involved in differential parasitism, cause and effect between TE density and DNA methylation level of these expanded gene families remain an open question and will be investigated thoroughly for future study.

ML of *Trichinella* can survive for long term in muscle cells, and ML can induce or reform the infected muscle cells into a niche to nurse them for survival, which is related with function of E/S products[30]. Here, we presented the landscape of transcriptomic changes between the two species and suggested aspects of differential immune system that might be associated with differential pathological characteristics. Several sets of E/S proteins that function at the host-parasite interface showed differential expression between the two species, such as C-type lectins, TP12446, and other parasitism-related genes like palmitoyl protein thioesterase (Supplementary Table 12), which may probably be associated with the establishment of enhanced parasite invasion and motility in *T. pseudospiralis* through higher expression of secreted proteins[31–33]. These genes may be of particular importance in *T. pseudospiralis* survival and/or adaption to the non-encapsulated phenotype, which represents a novel finding with respect to new parasite immune-evasion strategy in *Trichinella* spp., to the best of our knowledge. Intriguingly, here we observed the secreted proteins of TP12446 could induce inhibition of myotube formation and differentiation employing the C2C12 cell line. Similar results were also obtained from previous reports in C2C12 cells and primary myoblast cultures treated with muscle larvae excretory-secretory

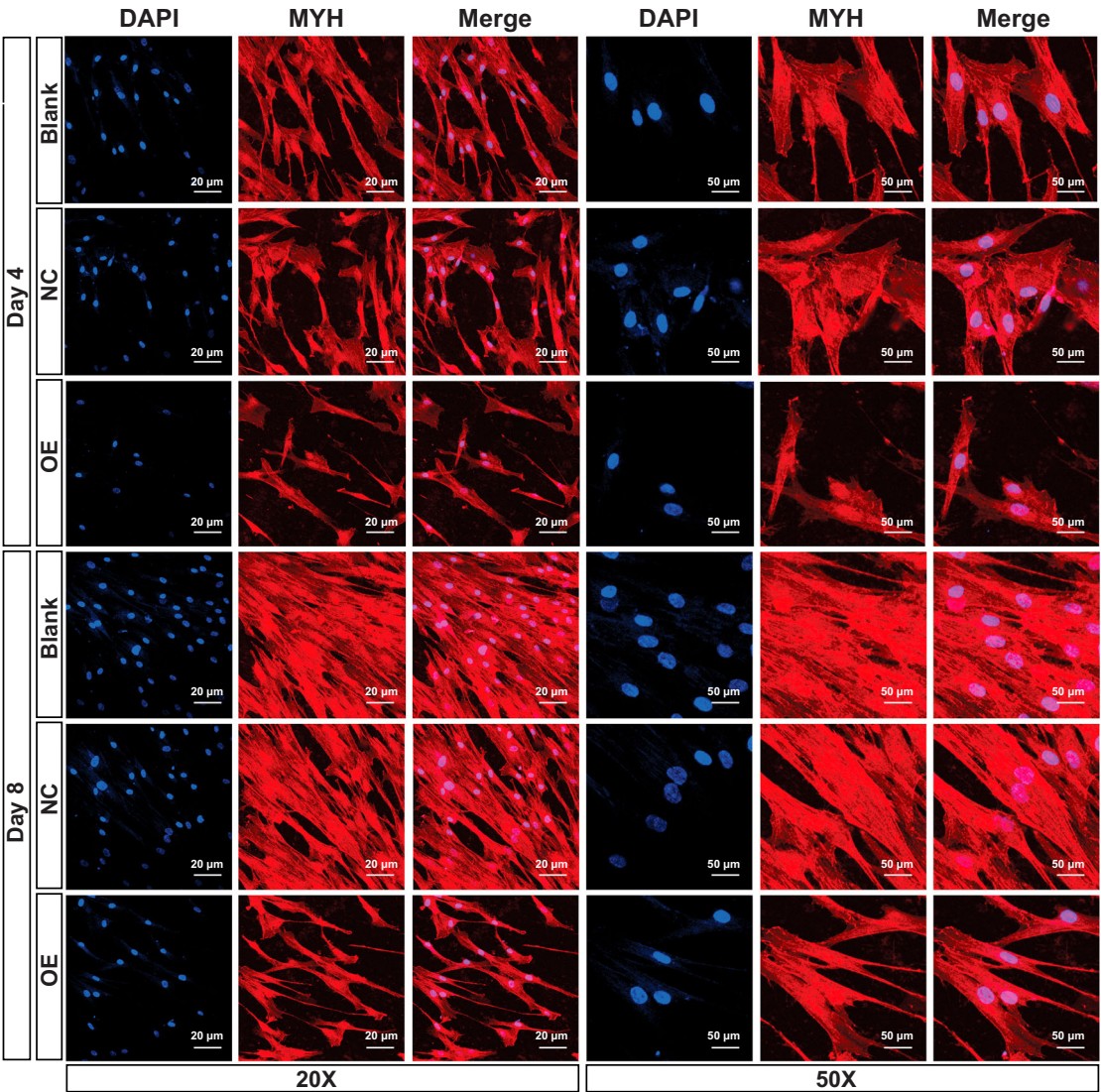

**Fig. 6 Secreted protein of TP12446 inhibits differentiation of C2C12 cells.** Cells were fixed after 4 or 8 days, stained with anti-Myosin Heavy Chain (MHC) antibody and counterstained with DAPI. The three transgenic mouse cell lines were: (1) normal cell (Blank), (2) an empty lentiviral vector PSE-CMV-NC (NC), (3) a lentiviral vector PSE-CMV-TP12446 with overexpression of TP12466 (OE). These images are representative of three biologically independent experiments.

products[34,35]. The induced cytoskeletal disarrangement may be a crucial step for muscle cell de-differentiation, which is a process occurred in muscle cell affected by *Trichinella* infection[36]. Moreover, TP12446 is released as a secreted active E3 ligase enzyme and exhibited homologous to mammalian RNF38 by ML stage in *T. pseudospiralis*. We speculated that it could affect the stability or function of a number of host proteins, including motor, sarcomeric and extracellular proteins, through protein ubiquitination as previous study demonstrated in *Trypanosoma cruzi*[31]. Thus, we will assess the whole impact of TP12446 on host muscle cell for future study.

In summary, our present study published a high-quality reference genome of non-encapsulated *T. pseudospiralis*. By comparing with the genome of encapsulated *T. spiralis* in genomics, epigenomics and transcriptomics, we found a great number of functional genes that are related with the unelucidated puzzle of differences in parasitism, pathology and immune response between the two clades of *Trichinella*. Our approaches used here will be applicable to other parasitic nematodes with public health and food safety importance.

## Methods

**Generation and maintenance of *T. pseudospiralis* inbred strains.** *Trichinella pseudospiralis* (ISS13), genotyped and proved by OIE Collaborating Center on Foodborne Parasites in Asian-Pacific Region, was preserved by serial passages in female BALB/c rats. Larvae were recovered from muscle tissue of infected rats on 35 days post infection (dpi) by artificial digestion with pepsin-HCl (1% pepsin, 1% HCl at 42 °C for 45 min). The Ad and NBL were isolated from the intestines of infected rats. Briefly, adult worms and newborn larvae were isolated from experimentally infected rats orally inoculated with *T. pseudospiralis* with a single dose of 8000 larvae per rat. Adult *T. pseudospiralis* worms were harvested by centrifugation of the fragmented intestines. Purified newborn larvae were harvested every 12 h from adult worms collected at 6 dpi and incubated in Iscove's Modified Dulbecco's Medium (IMDM) in 75-cm$^2$ cell culture plates at 37 °C. The animals were treated in strict according to the guidelines of the National Institutes of Health (NIH publication no. 85-23, revised 1996). All experimental protocol involving animals have been reviewed and approved by the Ethical Committee of the Jilin University affiliated with the Provincial Animal Health Committee, Jilin Province, China (Ethical Clearance number IZ-2009-08).

**DNA preparation and high-throughput sequencing.** High molecular weight genomic DNA were isolated from freshly collected muscle larvae using the phenol-chloroform extraction method. Total DNA amounts were determined using a Qubit Fluorometer dsDNA HS Kit (Invitrogen), according to the manufacturer's instructions. Genomic DNA integrity was verified by agarose gel electrophoresis

and using a BioAnalyzer (2100, Agilent). A paired-end library was constructed with an average insert size of 200 bp following the instructions provided by Illumina. Briefly, approximately 2000 nanograms of genomic DNA were fragmented by the Covaris S2 system. Then the fragmented DNA were blunt-ending, A-tailing and subsequently adapter-ligation. Sequencing was performed on the Illumina Hiseq 2500 platform. For PacBio sequencing, genomic DNA was shearing using the Covaris g-TUBE device as the manufacturer suggested. Then a 20-kb library was constructed for on a PacBio RS II platform (Pacific Biosciences). Both the Illumina sequencing reads and Pacbio SMRT sequencing data were used for the genome assembly.

**Genome assembly**. FastQC (v0.11.7) (https://github.com/s-andrews/FastQC) was used to assess the quality of raw sequencing reads. For Illumina reads, adapters and low-quality reads were filtered by Trimmomatic (v0.36)[37] (LEADING:3, TRAILING:3, SLIDINGWINDOW:4:15, MINLEN:75), approx. 6.90 billion high-quality data were generated. For PacBio reads, low-quality subreads (error rate > 0.2), short subreads (<5 kb), and duplicated reads were filtered to yield 9.93-billion of clean data. Illumina clean reads were assembled using Soapdenovo (v2.04)[38] with $k$-mer = 31. PacBio reads were assembled using Canu (v1.0)[39] algorithm, with the following parameters: error Rate = 0.01, genome Size = 70 Mb in addition to default parameters. One mate-paired library was downloaded from NCBI (PRJNA257433) based on pair-end relationships. Pilon (v1.22)[40] algorithm was then used to correct the Canu assembly using recommended settings. Then, the two initially assembled genomes were subjected to a process of merging and scaffolding with long-insert mate-paired reads by SSPACE (v3.0)[41]. Gap filling was performed by GapCloser (v1.12)[38] with the following parameters: GapCloser -a scafSeq -b gap_all.lib -o scafSeq.FG1 -t 10.

Several methods were used to assess the quality of the assemblies. BUSCO (v3.0)[42] was first used to assess completeness of T4_ISS13_R assembly based on the presence of single-copy orthologs from the OrthoDB database (www.orthodb.org). Further, expressed sequence tags (ESTs) downloaded from dbEST of GenBank (as of 08/15/2016) were used. A total of raw 56,575 reads (27,903,314 bp) were obtained. After removal of vector, poly-A, and low-complexity sequences, 54,485 reads (25,750,969 bp) were used for BLAT (https://github.com/djhshih/blat) analysis under default settings. We also mapped RNA-seq data from Ad, ML and NBL stages to the assembled genome using hisat2 (v2.0.2)[43] with default settings, achieving high mapping ratios ranging from 93.87% to 95.11%.

**Genome annotation**. Repeats were identified using a combination of de novo and homology-based approaches. Two de novo software packages, LTR_FINDER (v1.07)[44] and RepeatModeler (v1.0.11) (http://www.repeatmasker.org/RepeatModeler/), were used. Sequences with lengths >100 bp and gap 'N' < 5% constituted the repeat element libraries. Two software packages, including Tandem Repeats Finder (v4.09) and RepeatMasker (v4.0.7) (http://repeatmasker.org), were used in the homology-based prediction. TEs with 80% identity over 80% overlapping region were integrated together to construct a non-redundant repeat library. Transposon density of specific regions was calculated based on repeats belonging to known transposon classes.

Genes were predicted by a panel of combined approaches, including ab initio modeling, homologous gene prediction and transcript fragment mapping (including both ESTs and RNA-seq data). For ab initio-based method, three ab initio gene prediction softwares, including Augustus (v3.3.1)[45], Glimmer-HMM (v3.0.4)[46] and SNAP (v2006-07-28)[47], were applied on the genome with repeats >500 bp masked, except for miniature inverted-repeat TEs as those are usually located near genes or inside introns. For homologous-based prediction, the protein sequences of A. suum, B. malayi, C. elegans, T. spiralis, M. incognita, T. suis, H. sapiens, and Uniprot database were aligned onto T4_ISS13_R using TBLASTN with an E-value cutoff 1E-5 (http://www.pseudomonas.com/blast/settblastn), and the homologous regions were aligned against the matching proteins using GeneWise[48] to extract accurate exon and intron information. Evidence-based gene prediction was conducted using assembled transcripts by genome-referenced and de novo assemblies according to the same pipeline in homologous-based approach. Then the resulting transcript sequences were aligned to the T. pseudospiralis genome to determine exon and intron boundary information by PASA (v2.3.3)[49], which is a gene structure annotation and analysis tool. GLEAN[50] was used to integrate all the results to obtain a consensus gene set.

Annotation of the predicted genes of T. pseudospiralis was performed by aligning their sequences against a number of public nucleotide and protein sequence databases, including InterProScan (v54.0) (http://www.ebi.ac.uk/interpro/), Gene Ontology (GO) (http://geneontology.org/), Kyoto Encyclopedia of Genes and Genomes (KEGG) (59) (https://www.kegg.jp/), Swiss-Prot (release-2015_04), TrEMBL (release-2015_04), non-redundant (nr) (07/03/2015) by BLAST software with an E-value cutoff 1e−5. InterProScan was assessed for conserved protein domains. GO terms for each gene were obtained from the corresponding InterPro entry. The gene products were associated with a specific biochemical pathway using the KEGG database with an E-value filter of e-10.

The transfer RNA genes were identified by tRNAscan-SE (v1.3.1)[51] with default parameters. Then the ribosomal RNA genes were identified using RNAmmer (v1.2)[52]. Other noncoding RNAs, including miRNA, small nuclear RNA, were identified using INFERNAL (v1.1.1)[53] by sequence homology searches of the Rfam[54] database with default parameters.

Functional proteins and potential drug targets were annotated as follows: (1) peptidases and the proteins that inhibit them were predicted against the MEROPS (https://www.ebi.ac.uk/merops/) database; (2) GPCRs were searched against the Pfam database (http://pfam.xfam.org/) and known GPCRs in T. suis downloaded from NCBI using BlastP (E-value ≤ 1e−5). Proteins that predicted to have 3–15 predict transmembrane (TM) domains identified by phobius algorithm (v1.01)[55] were considered as GPCRs. 3) Kinase in T. suis were also downloaded from NCBI and used as queries to search against the proteomes using BlastP with E-value cutoff 1e−5. The identified domains were clustered with several species (H. sapiens, C. elegans, D. melanogaster) downloaded from KINBASE (http://kinase.com/kinbase/FastaFiles/) using OrthoMCL. Domains that failed to be assigned with other species were discarded for further analyses; (4) inferred E/S proteins were predicted using a strategy of integrating several tools. Proteins predicted to have a TM domain predicted by phobius algorithm were discarded for further E/S analysis. SignalP (v4.1)[56] was used to predict the presence of signal peptides from the first 70 N-terminal amino acids of each proteins (parameters for eukaryotes and default D-cutoff values). The two resulting hits were merged together to search homology by BlastP against Secreted Proteins Database (SPD) (ftp://ftp.cbi.pku.edu.cn/pub/database/spd/). Inferred secreted proteins had a signal peptide and lacked a recognizable transmembrane domain and showed BlastP homology to sequences in the SPD database. 5) Functional proteins identified above that showed homology against known drug target databases, including DrugBank (https://www.drugbank.ca/), ChEMBL (https://www.ebi.ac.uk/chembl/), Therapeutic Targets Database (http://bidd.nus.edu.sg/group/cjttd/), but exhibited no homology with the host H. sapiens proteomes with an E-value cutoff 1e−5 were considered as potential drug targets.

**WGBS library construction and data processing**. Two to five micrograms gDNA were sonicated to an approximately mean size range of 100–500 bp. After fragmentation, end-repair, addition of 3′ A bases and ligation of methylated cytosine PE adapters were performed, followed by bisulfite conversion of purified adapter-ligated DNA using an EZ DNA Methylation-Gold Kit™ (ZYMO Research, Irvine, CA, USA), according to the manufacturer's instructions. Size selection was achieved by PAGE gel and yielded DNA fragments of 250–450 bp from bisulfite conversion of purified adapter-ligated DNA. The converted DNA was then purified with the QIAquick Gel Extraction Kit, followed by PCR enrichment using JumpStart™ Taq DNA Polymerase for eleven cycles with Illumina PE PCR primers. Methylated-adapter ligated to unmethylated lambda-phage DNA (Promega, Madison, WI, USA) was used as an internal control for assessing the bisulfite conversion rate. Libraries were sequenced on Illumina HiSeq 2500 platform. Raw sequencing data were filtered for adapter contamination by cutadapt[57], parsed through quality filtration (quality cut off value = 5, low-quality rate < 0.5, 'N' rate < 0.1) and the trimmed reads shorter than 50 bp were discarded. Clean reads from each library were mapped to the updated genome, using BSMAP (v2.73)[58]. The methylation level of specific cytosine residues was estimated from the fraction of methylated sequence reads at that site (≥5× read depth). Mean methylation levels of specific genes or promoters were calculated by the read depth of methylated CpGs to the total sequenced depth of CpGs in that region.

**Transcriptome sequencing and differential expression**. Total RNA from T. pseudospiralis (Ad, NBL, and ML) was purified using Trizol reagent (Invitrogen, CA, USA), according to the manufacturer's instructions. RNA was dissolved in diethyl-pyrocarbonate (DEPC)-treated water and treated with DNase I (Invitrogen, CA, USA). The quantity and quality of the RNA were tested by ultraviolet-Vis spectrophotometry using a NanoDrop 2000 (Thermo Scientific CA, USA). Pair-end RNA-seq libraries were constructed, following Illumina's protocols, for the three life stages of T. pseudospiralis and sequenced on an Illumina HiSeq 2500 platform. Raw sequencing data were filtered for adapter contamination by cutadapt (http://code.google.com/p/cutadapt/), parsed through quality filtration using the program Trimmomatic (v0.36)[37] (Phred ≥ 20). TopHat (v2.0.12)[59] and followed by Cufflinks (v2.2.1)[60] to assemble the genome-referenced transcripts with default settings. Trinity (r20140717)[61] was used to conduct the de novo assembly. Gene expression analysis was measured using the reads per kb per million mapped reads (RPKM) and was calculated using Cufflinks. Differentially expressed genes were assessed with DESeq2 (http://www.bioconductor.org/packages/release/bioc/html/DESeq2.html).

**Gene family expansion and contraction analyses**. OrthoMCL[62] was used to confirm the genes that were orthologous among the following species: T. pseudospiralis, T. spiralis, B. malayi, C. elegans, M. incognita, D. melanogaster (outgroup). Single copy orthologous groups were extracted for phylogenetic tree construction using MrBayes (v3.2.2)[63] after aligning the family members with MUSCLE[64]. The optimal substitution models for amino acid and CDS sequences were estimated by ProtTest (v3.4.2)[65] and ModelTest (v0.1.0)[66], respectively. The divergence time for the six species using SCO gene families was estimated using the program MCMCTREE implemented in the PAML package[67]. The expansion or contraction events were determined using CAFÉ based on the comparison of orthologous gene family size differences under probabilistic graphical models[68].

**Domain feature prediction and protein structural modeling**. Conserved domains and transmembrane regions were identified using InterProScan and

SMART (http://smart.embl-heidelberg.de/smart/set_mode.cgi). DOG (Domain Graph, version 2.0)[69] was used to plot the domain architecture. I-TASSER (https://zhanglab.ccmb.med.umich.edu/I-TASSER/), which is a hierarchical approach to protein structure and function prediction, was used to predict protein 3D-structure from the PDB by multiple threading approach LOMETS (Local Meta-Threading Server). Structural alignment was performed by TM-align (https://zhanglab.ccmb.med.umich.edu/TM-align/), which is an algorithm for sequence independent protein structure comparisons. The representative models were visualized by PyMOL (https://pymol.org/2/).

**Cell culture and induction of differentiation**. C2C12 skeletal muscle cells were obtained from American Type Culture Collection (ATCC, Manassas, VA). Cells were cultured at 37 °C for 48 h under 5% $CO_2$ in Dulbecco's modified Eagle's medium-high glucose (DMEM, GIBCO, USA) supplemented with 10% (v/v) fetal bovine serum (FBS, GIBCO, USA), 100 U/ml penicillin and 100 μg/ml strepto-mycin. For differentiation into myotubes, cells were grown to 90% confluency. Cells was then switched to differentiation media at 37 °C for 4–5 d under 5% $CO_2$ in DMEM supplemented with 5% (v/v) horse serum (HS, GIBCO, USA), 100 U/ml penicillin and 100 μg/ml streptomycin.

**Vector construction, lentivirus preparation, and transduction**. Lentivirus vector used for overexpression of TP12446 was purchased from Sangon Biotech (Shanghai, China). The promoter of the backbone plasmid was replaced by cytomegalovious7 (CMV7) promoter at BamHI and XhoI restriction sites. cDNA sequence of TP12446 was amplified by PCR and subsequently integrated a 3XFLAG into the N-terminal of the cDNA using the following primers: forward primer (TP12446): TGGCAAA-GAATTGGATCCGCCACCATGTTCAGCTGGTCATTATCG, reverse primer (TP12446): AAAATCATGGGAGTTGCGTTG; forward primer (3XFLAG): CGCAACTCCCATGATTTTGACTACAAGGATGA, reverse primer (3XFLAG): CATAATACTAGTCTCGAGTTATTTGTCGTCATCATC. The resulting vector was confirmed by sequencing and designated PSE-CMV-TP12446. The lentivirus particles were made by co-transfection of HEK293T cells with expression vector PSE-CMV-TP12446 together with packaging plasmid pCMV-dR8.9 (Addgene 8455), the non-retroviral pCMV-VSV-G (Addgene 8454) plasmid and Opi-MEM (GIBCO, cat. No. 31985). Virus titer was determined using qPCR (ABI PRISM 7000) and the thermal cycling program was 50 °C for 2 min, 95 °C for 10 min, and then 40 cycles of 95 °C for 15 s and 60 °C for 1 min. The lentivirus supernatants were concentrated by ultra-centrifugation at 25,000 rpm (SW28 ultracentrifuge rotor, Beckman) for 2 h at 4 °C. For transduction, differentiated C2C12 cells were transfected with control lentivirus or lentivirus expressing TP12446 for 4 d and 8d at a multiplicity of infection (MOI) of 100. The transfection efficiency was confirmed using western blotting.

**Immuno-fluorescence analysis**. For immunostaining, C2C12 myoblasts grown on 6 well optical plates were washed in PBS 3× at days 0, 4, and 8 under differentiation conditions. Then the cultured cells were fixed with 4% (w/v) paraformaldehyde for 15 min at room temperature and followed by permeabilization with 0.1% Triton-100 diluted in PBS for 15 min at room temperature. The cells were blocked with 3% (w/v) BSA in PBS for 1 h at 37 °C and incubated with monoclonal mouse anti-myosin heavy chain (MYH1/2/3, Santa Cruz Biotechnology, USA) diluted in 1% BSA-PBT at 4 °C overnight. Cells were then incubated with a secondary antibody (Goat Anti-Rabbit IgG H&L, Alexa Fluor® 555, ab150078) diluted at 1:1000 for 1 h at 37 °C and washed in PBS 3× for 10 min. For the nuclei staining, the cells were incubated with DAPI (1:1000) for 10 min at room temperature. Finally, the cells were visualized with laser scanning confocal microscopy (Zeiss, LSM880).

**Statistics and reproducibility**. The detection of enzymatic activity of DNMTs, transfection efficiency of TP12446 into lentivirus vector, and immune-fluorescence analysis of C2C12 myoblasts were conducted with at least two biologically inde-pendent replicates. Methylation levels between different genomic contexts were compared using the Mann–Whitney $U$ test, followed by multiple Benjamini–Hochberg testing corrected $P < 0.05$. Gene expression levels between different species or stages were compared by Student's $t$-test with Benjamini–Hochberg corrected $P < 0.05$. All the statistical analyses were done using available packages in R (version 3.6.2).

**Reporting summary**. Further information on research design is available in the Nature Research Reporting Summary linked to this article.

## Data availability

Raw genome sequencing data were deposited in the National Center for Biotechnology Information with the following accession number: SAMN08905168 under project PRJNA451013; SRP140458 for transcriptome and WGBS data; The genome sequence has also been deposited at DDBJ/ENA/GenBank (accession number QAWF00000000).

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

## Acknowledgements

We thank William J Lucas and Zhiliang Wu for help in editing the manuscript for grammar and writing style. We thank Jue Ruan, Peng Cui, and Qiang Lin for helpful guidance on bioinformatic analysis. This study was supported by the National Natural Science Foundation of China (31520103916, 31872467), the National Key Research and Development Program of China (2017YFD0501302, 2017YFC1601206, 2016YFD0500707, 2018YFC1602504), Guangdong Innovative and Entrepreneurial Research Team Program (No.2014ZT05S123), Program for JLU Science and Technology Innovative Research Team, The Agricultural Science and Technology Innovation Program and The Elite Young Scientists Program of CAAS.

## Author contributions

Fei Gao and Mingyuan Liu conceived the project. Xiaolei Liu and Bin Tang undertook sample collection. Xue Bai and Xuelin Wang performed DNA extraction, and Xiaolei Liu and Yong Yang performed RNA extraction experiments. Rui Qin and Xinxin Yu undertook WGBS and RNA-seq library construction. Xiaolei Liu performed the construction of lentivirus vector and immune-fluorescence analysis experiments. Yayan Feng carried out all of the bioinformatic analysis. Yayan Feng, Xiaolei Liu, and Fei Gao wrote the manuscript. All authors have read and approved the manuscript for publication.

## Competing interests

The authors declare no competing interests.
