## [Peer Review File · Communications Biology]

Reviewers' comments:

Reviewer #1 (Remarks to the Author):

The manuscript entitled "Molecular basis of differential 1 parasitism between non-encapsulated and encapsulated *Trichinella* revealed by a high-quality genome assembly" by Liu et al is a well thought and well-written manuscript. I have a few concerns as mentioned below.

Concerns:

26/67/95: if you the title says "high-quality genome assembly" then why the encapsulated genome was not sequenced? How do you know that encapsulated genome is a high-quality genome—what analysis did you perform to confirm it? I know that the authors mentioned in their manuscript but how did you confirm it.

52: spp should not be italicized—correct it all over the manuscript.

99-111: Be clear—confusing.

112: write the depth of repetitive region; it is important to understand—how was the coverage of different regions of the genome.

115-117: I would like to see more details of this repetitive region before making this conclusive statement.

119: Previously, we....

184: Indicate the basis of this estimation—the information is not sufficient for this statement.

192: no background about these new species in the introduction—where is the phylogenetic tree or analysis indicating this? Reference? Are these the only phylogenetically related species? What was the cutoff to consider only these species as phylogenetically related? How did you generate the gene families? Did you sequence these or retrieved from Database? Clearly indicate this all the information here.

216 delete "then"

223: What was the pattern among other gene families other than the DNaseII family? It would be great if authors can explain one more gene family (with the same criteria) to understand if the pattern is the same among the gene families.

286-287: how do you know that TEs and DNA methylation played important roles in differential parasitism? Did you experimentally prove it? Or someone else?

412: what kind of recombinant?

418-19: provide proper reference to follow the protocol

426: what method/kit was used to isolate the DNA

425-434: provide more information about sequencing libraries and kits used with a brief description

435: what was the depth for each sequencing data—supposed to be very high as you claimed; provide the details here.

Reviewer #2 (Remarks to the Author, please see attached with email):

Reviewer #3 (Remarks to the Author):

The manuscript by Gao et al, presents a high quality de novo assembled genome of *T. pseudospiralis*. Together with multi-omics data, comparative analysis revealed the potential virulence factors and mechanisms shaping the parasitism-related gene family expansion. The manuscript is generally well written, the data analysis workflow is well constructed, and statistical tests were done properly. Figures and tables demonstrated clear and detailed support information. However, authors should provide a few clarifications to make the arguments stronger and clearer.

1. In line 41, remove one of the two "roles".
2. In line 88, N50 is 208.90 Kb, which is equal to Table 1, however, the number 68 is not the same as shown 65 in Table 1. Please confirm the number is accurate.
3. In line 92, BUSCO was used to assess completeness but marked as CEGMA in Table 1. CEGMA has been discontinued since 2015. Please correct.
4. In line 94, 88.9% of core genes were present and complete, this is inconsistent to 95.56% in Table 1. Please verify.
5. In line 97, the new assembly was named T4_ISS13_R, but in Supplementary Method, T4_ISS13_r2.0 was used. If T4_ISS13_r2.0 is T4_ISS13_R please correct it.
6. In line 109, "Most of these proteins had the support of gene expression data". Can we clarify to what percentage of these proteins had expression data support?
7. In line 234, ~60% of expanded genes are tandem duplication. Just curious how many of them are in the same direction and opposite direction? It may offer a clue of the major duplication mechanism.
8. In line 249, the Spearman correlation coefficients were calculated. Is it possible to show the P-value based on the number of samples tested? APA format is $r(27) = 0.558, p < 0.001$. The number following r in parentheses corresponds to the sample size.
9. In line 444, "5 mate-pair library data were downloaded from NCBI (PRJNA257433)". Can we specify what are the 5 datasets? There are only 4 datasets for *Trichinella pseudospiralis* (ISS13), and only 1 of them is mate paired.

From the Reviewer #1:

Q1. 26/67/95: if you the title says “high-quality genome assembly” then why the encapsulated genome was not sequenced? How do you know that encapsulated genome is a high-quality genome—what analysis did you perform to confirm it? I know that the authors mentioned in their manuscript but how did you confirm it.

Answer:

The *T. spiralis* genome was published and fully evaluated for its quality¹. Further we performed BUSCO analysis to assess completeness of the two *Trichinella* genomes in our study. We revealed that the two genomes exhibited comparable percentage of single-copy orthologs genes from the OrthoDB database (88.6% in *T. pseudospiralis* versus 89.2% in *T. spiralis*). Therefore, we confirmed that the two *Trichinella* genome assemblies were both with high-quality and comparable with each other. We now added the corresponding text in **line 105-111** to the main text and corrected the corresponding information in Table 1.

Despite that, indeed the previous title was not clear, so we use a new title “Comparative multi-omics analyses reveal molecular basis of differential parasitism between non-encapsulated and encapsulated *Trichinella*” in the revised manuscript.

Q2. 52: spp should not be italicized—correct it all over the manuscript.

Answer: Thank you for this advice. We have now corrected this all over the manuscript.

Q3. 99-111: Be clear—confusing.

Answer: Thank you for this advice. We further revised this paragraph in **line 122-135**, with the genome version clearly indicated and “approx. 93.0%~95.6% of these proteins had support of gene expression data” indicated.

Q4. 112: write the depth of repetitive region; it is important to understand—how was the coverage of different regions of the genome.

Answer: Thank you for this advice. The coverage folds of each repeat region in the *T. pseudospiralis* genome were estimated by bedtools (v2.29.2). Then we plotted the density distribution of read depth of each repeat region and revealed that approx. 86.5% of these repeat regions have high coverage folds ($\geq 150x$) (**Figure R1**), significantly higher than the read depth of genome overall ($\sim 98x$). We have now added the corresponding text in **line 114-115** to the main text.

Figure R1. Read depth distribution in repeat regions of the *T. pseudospiralis* genome.

Q5. 115-117: I would like to see more details of this repetitive region before making this conclusive statement.

Answer: Thank you for this advice. We have added more analyses between the two assembly versions of *T. pseudospiralis* both at gene features and repeat contents. We found that the two assembly versions exhibited comparable gene features, such as gene number, exon size, intron size and coding region size. However, the T4_ISS13_R also had approx. 18.9 Mb more repeat sequences than T4_ISS13_r1.0, accounting for most of the differences in genome size between the two assembly versions. We have now added Table S1 to the Supplementary material and added the text in **line 115-120** to the main text.

Table S1. Comparison of genome features between the two assembly versions of *T. pseudospiralis* genome.

Term	T4_ISS13_R	T4_ISS13_r1.0
Overall		
17-mer estimated genome size (in Mb)	69.79	69.78
Total scaffolded assembly size (Mb); total scaffolds	68.90; 2,746	49.16; 7,221
Total scaffolds of > 2kb: length (Mb); no. of scaffold	68.71; 2,631	46.64; 406
Largest scaffold (Mb)	3.73	1.95
Gaps, combined length (kb)	29.69	905.3
N50 in kb of scaffolds; count > N50 length	208.90; 68	235.43; 51
N90 in kb of scaffolds; count > N90 length	7.68; 1,231	60.44; 206
N50 in kb of contigs; count > N50 length	174.42; 75	122.06; 112
N90 in kb of contigs; count > N90 length	7.56; 1,321	13.81; 415
Repetitive sequences (%)	40.28%	18.01%
BUSCO genes (%)	88.6%	87.2%
Protein coding regions		
Number of gene models	12,682	12,656
Gene density (genes per Mb)	184	257
Exonic proportion, including introns (bp)	29,554,476	28,674,747

Mean/Median gene size (bp)	2330.43/1077	2265.70/1089
Mean/Median CDS size (bp)	1039.25/627	1045.78/523.5
Number of exons	69,966	76,053
Number of bp included in exons	13,179,752	13,235,355
Mean/Median exon size (bp)	188.37/137	174.03/124
Mean number of exons per gene	5.52	6.01
Number of introns	57,284	63,397
Number of bp included in introns	16,374,723	15,439,393
Mean/Median intron size (bp)	285.85/76	243.54/76
Overall G + C content (%)	31.49	32.60
Exons, G + C content (%)	42.25	42.33
Introns, G + C content (%)	28.90	27.45
Intergenic regions, G + C content (%)	29.48	30.32
Number of predicted excretory-secretory protein	471	443
Number of expressed genes (RPKM > 10)	7,170	5,874

Q6. 119: Previously, we....

Answer: Thank you for this advice. We have corrected this in the manuscript.

Q7. 184: Indicate the basis of this estimation—the information is not sufficient for this statement.

Answer: Thank you for this advice. In this part, we firstly estimated the divergence of the encapsulated and non-encapsulated *Trichinella*, an event estimated to have occurred 22.6 (15.3~28.1) million years ago (Supplementary Figure 10). Then we calculated insertion times of these LTRs. We discovered that a burst of LTR activity likely occurred during the last five million years (Supplementary Figure 9). We have changed the text in **line 190-195** in the manuscript.

Q8. 192: no background about these new species in the introduction—where is the phylogenetic tree or analysis indicating this? Reference? Are these the only phylogenetically related species? What was the cutoff to consider only these species as phylogenetically related? How did you generate the gene families? Did you sequence these or retrieved from Database? Clearly indicate this all the information here.

Answer:

Thank you for this important comment. In the part of gene family expansion and contraction analysis, we selected species from four major phylogenetically-related lineages that collectively span the phylum, including two *Trichinella* species (*T. spiralis* and *T. pseudospiralis*), one non-parasite *C. elegans*, one plant parasite, *M. incognita*, one animal parasite, *B. malayi*, thus representing diverse trophic ecologies. Arthropod (*D. melanogaster*) was used as outgroup. Thus, we selected the phylogenetically related species for comparative genomic analysis. We have now added all the information in **line 200-204** to the main text.

Then we used OrthoMCL to confirm the genes that were orthologous among these species. The expansion or contraction events were determined using CAFÉ based on the comparison of orthologous gene family size differences under probabilistic graphical models. These texts are presented in **line 536-545** to the main text.

Q9. 216 delete “then”

Answer: We have now deleted ‘then’ from the manuscript.

Q10. 223: What was the pattern among other gene families other than the DNaseII family? It would be great if authors can explain one more gene family (with the same criteria) to understand if the pattern is the same among the gene families.

Answer: Thank you for this advice. In the part ‘Differential expansion of gene families between the two clades *Trichinella*’, apart from the DNase II genes, we have now added another gene family ‘Protein of unknown function DUF1759’ in **line 232-235** in the main text to understand if the pattern is the same among the gene families.

Q11. 286-287: how do you know that TEs and DNA methylation played important roles in differential parasitism? Did you experimentally prove it? Or someone else?

Answer: Thank you for this important comment. Indeed, in this manuscript, we just observed several expanded gene families with increased levels of TE density and DNA methylation. Cause or effect between TEs and DNA methylation and gene family expansion were not further studied. Therefore, we revised the manuscript with regards to the association between TE density and DNA methylation across the whole manuscript in **line 382-394**.

Q12. 412: what kind of recombinant?

Answer: Thank you for this important comment. The strains were preserved by serial passages in BALB/c rats. We have revised the text in **line 429-439**.

Q13. 418-19: provide proper reference to follow the protocol.

Answer: We have added more details about sample collection and maintenance in **line 429-443**.

Q14. 426: what method/kit was used to isolate the DNA.

Answer: The DNA were isolated from muscle larvae using the phenol chloroform extraction method. We have added the information in **line 446-450** to the main text.

Q15. 425-434: provide more information about sequencing libraries and kits used with a brief description.

Answer: Thank you for this advice. We have added a brief description of Illumina and PacBio sequencing kits in **line 447-458** in the manuscript.

Q16. 435: what was the depth for each sequencing data—supposed to be very high as you claimed; provide the details here.

Answer: Thank you for this advice. The average read depth of PacBio and Illumina reads were 144x and 98x, respectively. This information has presented in Supplementary Table 1. We have now added this information in **line 97-98** to the manuscript.

From the Reviewer #2:

This manuscript concerns the gene content of *Trichinella* genome sequences. A comparison is made between the existing *T. spiralis* (Ts) genome sequence and a new sequence for *T. pseudospiralis* (Tp). The study observes that 1) there are more repetitive elements in the Tp sequence; 2) there have been independent expansions of known multi-copy gene families in both species; 3) the degree of base methylation is greater for these expanded gene families than non-expanded genes in both species; and 4) base methylation is higher in Tp than Ts.

This leads to the conclusion that multiplication of repetitive elements within both genomes has driven divergence in parasite specific effector gene repertoire and expression, through some undefined interplay between repetitive elements, methylation and gene duplication, and that this explains the phenotypic differences between the species.

The sequencing and analysis are well done, and I have no issues with this. I do criticise the writing of the manuscript and the author's interpretation of this data. Overall, I think this is a competent, publishable analysis that will benefit workers in the field and is an incremental contribution to our understanding of how genomes evolve in concert with transposable elements. The genome appears to be a good quality resource. It may influence the field to show that the genomic trends highlighted here are true across the clade and hold up in population comparisons, others may functionally assay these species-specific proteins and demonstrate their role in infection phenotypes, species-specific or not.

Major points:

Q1. Having read the manuscript multiple times, I find it difficult to understand how the data support the interpretation. Part of this problem is that the phenotypic question being asked is not adequately and accurately defined. The title says that the molecular basis for differential parasitism is reported. What is the difference between the two organisms being compared at the genomic level? One is encapsulated, one not. What does this mean and why does it matter? Pathological differences between the clades are eluded to (l157-59) but not defined in detail. How are we supposed to evaluate if

the genetic differences observed can possibly explain the phenotypic differences between the organisms? What is the comparison trying to explain?

Answer:

Thank you for raising these concerns. Indeed, the phenotypic differences between the two clades *Trichinella* were not clearly defined in our manuscript. Thus, we added more description in the background (**line 41-68**) of the manuscript to point out that difference between the two clades *Trichinella* in parasitological, pathological and immunological characteristics.

Further, we performed comparative multi-omics analysis between the two clades *Trichinella* and revealed differential expansion and methylation of parasitism-related multi-copy gene families, especially for DNase II genes and GSTs. We indeed cannot provide evidence to confirm the cause or effect between the TE density and DNA methylation based on current results, just revealed that the expanded gene families exhibited increased TE content and DNA methylation level. Importantly, we revealed divergent key E/S genes between the two species, especially for TP12446, which could induce inhibition of myotube formation and differentiation, which could possibly explain the phenotypic differences and further provide molecular basis of differential parasitism between the two clades *Trichinella*.

Q2. A second problem with the interpretation relates to cause and effect. How do the authors know that transposable elements have driven gene family expansion? It could be equally possible that, having duplicated, these genes became targets for element insertion because they are functionally redundant, and so not subject to the strong purifying selection protecting core genes. I think the authors should recognize that both possibilities are equally possible (the observed methylation patterns are irrelevant) in the abstract and in the text, e.g. at l210-2. The narrative seems to be inductive – as if the authors have found the greatest difference between the genomes and built an explanation around that difference, rather than precisely defining the phenotypic difference, raising a hypothesis of what could explain this, and then testing that. Regardless, there is only correlative, circumstantial evidence for repetitive elements causing gene duplication.

Answer: Thank you for this important comment. We indeed agree that we cannot confirm the cause or effect between the TE density and DNA methylation based on current results, we just revealed that the expanded gene families exhibited increased TE content. Thus, we revised all over the manuscript just to point out associations between TE and gene family expansion.

Q3. The authors might also reflect on whether the observed methylation of these multi-copy genes might reflect their own regulation, rather than, or in addition to, transposable elements within them. It is common for such families to display phasic expression, whereby individual paralogs are switched on and off rapidly, creating

highly idiosyncratic expression profiles. This is typically achieved by regulating the chromatin structure around their (typically sub-telomeric) loci. An example of such genes including transposable elements might be the Retrotransposon hot spot proteins in African trypanosomes.

Answer: Thank you for this important suggestion. We revised all over the manuscript accordingly in **line 388-394**.

Q4. With regard to the gene family expansions, these seem to have happened on a broadly similar magnitude in both genomes (median comparison). In other parasite genomes, it is commonplace for such paralogous gene families to increase and decrease in size among closely related species, it is well understood that they have rapid turnover of members resulting in divergence, mutual exclusivity of repertoire and widespread copy number variation. There is nothing in the study to warrant a claim that these expansions explain differences between the species, they are just coincident with the distinct phenotypes. Without any functional assays on species-specific genes, I do not see how the study has established the “molecular basis for differential parasitism” (whatever that differential actually is). I would challenge the authors to answer one question: what is it among your data that explains why Tp is non-encapsulated while Ts is encapsulated (and everything that might involve phenotypically)?

Answer:

Thank you for this important question. To reveal molecular basis of differential parasitism between the two clades *Trichinella*, we performed comparative multi-omics analysis, including genomics, epigenomics and transcriptomics. We revealed that parasitism-related multi-copy gene families, especially for DNase II genes and GSTs, exhibited differential expansion and methylation.

Furthermore, methylome and transcriptome analyses revealed divergent key excretory/secretory (E/S) genes between the two clades *Trichinella*. Among these key E/S genes, TP12446 is significantly more expressed across three life stages in *T. pseudospiralis*. Overexpression of TP12446 in the mouse C2C12 skeletal muscle cell line could induce inhibition of myotube formation and differentiation (**Figure R2**), further indicate its key role in parasitism of *T. pseudospiralis*. We have now added these results in the manuscript in **line 314-336** (Results) and **line 559-603** (Methods).

Figure R2. Secreted protein of TP12446 inhibits differentiation of C2C12 cells. Cells were fixed after 4 or 8 days, stained with anti-Myosin Heavy Chain (MHC) antibody and counterstained with DAPI. The three transgenic mouse cell lines were: 1) normal cell (Blank), 2) an empty lentiviral vector PSE-CMV-NC (NC), 3) a lentiviral vector PSE-CMV-TP12446 with overexpression of TP12466 (OE). These images are representative of three independent experiments.

Q5. Spelling and grammar is routinely below acceptable standards. There are too many instances to offer corrections. The entire manuscript needs to be checked for such errors.

Answer: Thanks for this advice. We further thoroughly revised the manuscript.

From the Reviewer #3 (Remarks to the Author):

The manuscript by Gao et al, presents a high quality de novo assembled genome of *T. pseudospiralis*. Together with multi-omics data, comparative analysis revealed the potential virulence factors and mechanisms shaping the parasitism-related gene family expansion.

The manuscript is generally well written, the data analysis workflow is well constructed, and statistical tests were done properly. Figures and tables demonstrated clear and detailed support information. However, authors should provide a few clarifications to make the arguments stronger and clearer.

Q1. In line 41, remove one of the two "roles".

Answer: Thanks for this advice. We have re-written the background of the manuscript and have corrected this mistake.

Q2. In line 88, N50 is 208.90 Kb, which is equal to Table 1, however, the number 68 is not the same as shown 65 in Table 1. Please confirm the number is accurate.

Answer: Thanks for this advice. We checked the manuscript thoroughly and corrected all the discrepancies between the text and Table 1 across the manuscript.

Q3. In line 92, BUSCO was used to assess completeness but marked as CEGMA in Table 1. CEGMA has been discontinued since 2015. Please correct.

Answer: Thanks for this advice. We checked the manuscript thoroughly and corrected the discrepancies between the text and Table 1.

Q4. In line 94, 88.9% of core genes were present and complete, this is inconsistent to 95.56% in Table 1. Please verify.

Answer: Thanks for this advice. We used BUSCO to assess the genome completeness and corrected the corresponding information in Table 1.

Q5. In line 97, the new assembly was named T4_ISS13_R, but in Supplementary Method, T4_ISS13_r2.0 was used. If T4_ISS13_r2.0 is T4_ISS13_R please correct it.

Answer: We feel sorry for the discrepancies in supplementary material. The newly generated genome assembly version of *T. pseudospiralis* is named T4_ISS13_R. Thus, we changed all the T4_ISS13_r2.0 to T4_ISS13_R in supplementary file.

Q6. In line 109, "Most of these proteins had the support of gene expression data". Can we clarify to what percentage of these proteins had expression data support?

Answer: We calculated number of functional proteins that had support of gene expression data and observed that approx. 93.0%~95.6% of these proteins had support of gene expression data. We have already added these results in **line 133-135** in the main text.

Q7. In line 234, ~60% of expanded genes are tandem duplication. Just curious how many of them are in the same direction and opposite direction? It may offer a clue of the major duplication mechanism.

Answer: Thanks for this advice. In present study, we performed comparative multi-omics analysis between the two clades *Trichinella* and revealed differential expansion and methylation of parasitism-related multi-copy gene families, especially for DNase II genes and GSTs. We indeed cannot provide evidence to confirm the cause or effect between the TE density and DNA methylation based on current results, just revealed that the expanded gene families exhibited increased TE content and DNA methylation level. Importantly, we revealed divergent key E/S genes between the two species, especially for TP12446, which could induce inhibition of myotube formation and differentiation, which could possibly explain the phenotypic differences and further provide molecular basis of differential parasitism between the two clades *Trichinella*. Following this framework, the gene duplication pattern appeared to irrelevant with our current study. Thus, we removed the corresponding results from the main text.

Q8. In line 249, the Spearman correlation coefficients were calculated. Is it possible to show the P-value based on the number of samples tested? APA format is “ $r(27) = 0.558, p < 0.001$ ”. The number following r in parentheses corresponds to the sample size.

Answer: Thanks for this advice. As mentioned above, it appeared that gene dosage rebalance was irrelevant with our current study. Thus, we removed this part from the manuscript.

Q9. In line 444, “5 mate-pair library data were downloaded from NCBI (PRJNA257433)”. Can we specify what are the 5 datasets? There are only 4 datasets for *Trichinella pseudospiralis* (ISS13), and only 1 of them is mate paired.

Answer: We feel sorry for the mistakes in our main text. We just downloaded one mate-pair libraries used for genome assembly from NCBI. We have corrected the information in **line 468-469**.

References

- 1 Mitreva, M. *et al.* The draft genome of the parasitic nematode *Trichinella spiralis*. *Nat Genet* **43**, 228-235, doi:10.1038/ng.769 (2011).

REVIEWERS' COMMENTS:

Reviewer #2 (Remarks to the Author):

The written language of the manuscript is improved and now adequate. Some effort has been made to explain the phenotypic differences between the species, and to link differentially expressed features to phenotype. I still dislike the title. I think it exaggerates their conclusions, since I do not believe that they have identified 'the molecular basis of differential parasitism' precisely, but that is a matter of degree.

Response to referees for the decision on manuscript **COMMSBIO-20-0032**

From the Reviewer #2:

Q1. The written language of the manuscript is improved and now adequate. Some effort has been made to explain the phenotypic differences between the species, and to link differentially expressed features to phenotype. I still dislike the title. I think it exaggerates their conclusions, since I do not believe that they have identified 'the molecular basis of differential parasitism' precisely, but that is a matter of degree.

Answer:

Thank you for this advice. We have now changed our title as 'Comparative multi-omics analyses reveal key genes of differential parasitism between non-encapsulated and encapsulated *Trichinella*'.